# Association of Low Handgrip Strength with Chemotherapy Toxicity in Digestive Cancer Patients: A Comprehensive Observational Cohort Study (FIGHTDIGOTOX)

**DOI:** 10.3390/nu14214448

**Published:** 2022-10-22

**Authors:** Pierre Martin, Damien Botsen, Mathias Brugel, Eric Bertin, Claire Carlier, Rachid Mahmoudi, Florian Slimano, Marine Perrier, Olivier Bouché

**Affiliations:** 1Department of Medical Oncology, Godinot Cancer Institute, 51100 Reims, France; 2Department of Gastroenterology and Digestive Oncology, Université de Reims Champagne-Ardenne, CHU Reims, 51100 Reims, France; 3Department of Nutrition, Endocrinology and Diabetology, CHU Reims, 51100 Reims, France; 4Department of Internal Medicine and Geriatrics, Université de Reims Champagne-Ardenne, VieFra, CHU Reims, 51100 Reims, France; 5Department of Pharmacy, Université de Reims Champagne-Ardenne, CHU Reims, 51100 Reims, France

**Keywords:** digestive system neoplasms, dose-limiting toxicity, dynapenia, muscle strength, sarcopenia, frailty, clinical nutrition, malnutrition

## Abstract

In the FIGHTDIGO study, digestive cancer patients with dynapenia experienced more chemotherapy-induced neurotoxicities. FIGHTDIGOTOX aimed to evaluate the relationship between pre-therapeutic handgrip strength (HGS) and chemotherapy-induced dose-limiting toxicity (DLT) or all-grade toxicity in digestive cancer patients. HGS measurement was performed with a Jamar dynamometer. Dynapenia was defined according to EWGSOP2 criteria (<27 kg (men); <16 kg (women)). DLT was defined as any toxicity leading to dose reduction, treatment delay, or permanent discontinuation. We also performed an exploratory analysis in patients below the included population’s median HGS. A total of 244 patients were included. According to EWGSOP2 criteria, 23 patients had pre-therapeutic dynapenia (9.4%). With our exploratory median-based threshold (34 kg for men; 22 kg for women), 107 patients were dynapenic (43.8%). For each threshold, dynapenia was not an independent predictive factor of overall DLT and neurotoxicity. Dynapenic patients according to EWGSOP2 definition experienced more hand-foot syndrome (*p* = 0.007). Low HGS according to our exploratory threshold was associated with more all-grade asthenia (*p* = 0.014), anemia (*p* = 0.006), and asthenia with DLT (*p* = 0.029). Pre-therapeutic dynapenia was not a predictive factor for overall DLT and neurotoxicity in digestive cancer patients but could be a predictive factor of chemotherapy-induced anemia and asthenia. There is a need to better define the threshold of dynapenia in cancer patients.

## 1. Introduction

Digestive cancers are among the most common spectrum of cancer in the world [1]. Anticancer agents have potential acute and chronic toxicities which may require treatment dose adaptations. Identifying predictive factors could help physicians to prevent the occurrence of chemotherapy-induced dose-limiting toxicity (DLT).

Low lean body mass and sarcopenia have been shown to predict anticancer drug toxicity in patients with breast or colorectal cancer [2,3,4]. Sarcopenia was primarily defined as the age-related progressive and generalized loss of skeletal muscle mass [5]. The European Working Group on Sarcopenia in Older People (EWGSOP) extended the definition of sarcopenia as the association of low muscle mass, plus low muscle strength or low physical performance, occurring in various diseases [6]. Many studies have evaluated sarcopenia using measurements of muscle mass quantity or quality with whole-body imaging methods, especially computed tomography [7]. Nevertheless, these methods are costly, time-consuming, irradiating, and not adapted for routine clinical practice. In 2019, EWGSOP2 recommended using the handgrip strength (HGS) measurement to screen sarcopenia [5]. Loss of muscle strength, also named dynapenia [8], has been defined by EWGSOP2 consensus as HGS < 27 kg in men and <16 kg in women, based on the geriatric part of a cohort study [5,9].

Nevertheless, a great heterogeneity of cut-off points is presented in the literature, and not adapted for cancer patients [9,10,11]. HGS has already proven its interest in the elderly, since the loss of HGS has been associated with more postoperative complications, increased length of hospitalization, higher rehospitalization rate, and poorer physical status [12,13]. In cancer patients, dynapenia has been associated with cancer-related fatigue [14], poor quality of life [15], postoperative complications [16], and mortality [17]. HGS could also be a reliable and effective tool to screen for malnutrition in digestive cancer patients [18].

The FIGHTDIGO study has demonstrated the feasibility and acceptability of HGS measurement using a JAMAR dynamometer in an outpatient cancer unit [19]. An ancillary analysis from a small sample of FIGHTDIGO patients suggested that patients with pre-therapeutic dynapenia (defined by using cut-off points of <30 kg in men and <20 kg in women) experienced more chemotherapy-induced dose-limiting neurotoxicity (DLN), but no difference in terms of other DLT [20]. Considering the new HGS cut-off points recommended by EWGSOP2 to define sarcopenia, additional studies are required to confirm dynapenia as a potential predictor of DLN or DLT.

The present FIGHTDIGOTOX study aimed to assess the relationship between pre-therapeutic HGS and chemotherapy-induced DLT and/or all-grade toxicity in digestive cancer patients treated in an outpatient cancer unit.

## 2. Materials and Methods

### 2.1. Design and Population

FIGHTDIGOTOX is a comprehensive observational retrospective monocentric cohort study including patients older than 18 years old, diagnosed with primary digestive cancer and receiving an intravenous anticancer drug in the Oncology Day-Hospital of the Reims university hospital in France. From November 2015 to December 2018, patients aged more than 65 years-old had an HGS measurement before initiation of chemotherapy in the prospective AgElOn study (NCT02807129). From November 2018 to March 2020, each newly admitted patient (waiting for a first anticancer drug infusion) was invited to perform HGS measurement as part of the routine practice.

Patients were excluded if they had a history of previous anti-cancer treatment, did not understand, or practice the HGS test, had any history of neuromuscular disorder, had received exclusive oral chemotherapy or immunotherapy, and/or had early stopped anticancer treatment (≤1 cycle) unrelated to adverse effects.

### 2.2. Outcomes

The primary objective was to study the association between pre-therapeutic dynapenia with chemotherapy induced all-grade toxicities and DLT. The secondary objective was to analyze the same association using an exploratory median-based HGS threshold to define dynapenia.

### 2.3. Ethical Approval

The study was conducted in accordance with the Helsinki Declaration. Informed written consent was obtained from each patient enrolled in the AgElOn trial. This trial was approved by the ethics committee (Committee for the Protection of Person EST I DIJON, 25 March 2016) and was registered on Clinicaltrials.gov (NCT02807129). Patients’ records were anonymized prior to analysis. The database was constituted in accordance with the reference methodology MR004 of the French National Commission on Informatics and Liberty (CNIL). As per French regulations concerning the retrospective study, no informed consent or additional ethical committee review was required.

### 2.4. Data Collection

Patients’ characteristics of interest (including sex, age, tumor location, disease stage, comorbidities, anticancer drug regimen, concomitant radiotherapy, ECOG Performance Status (PS), Body Mass Index (BMI), G8 score in older patients (tool to identify elderly cancer patients who benefit from a comprehensive geriatric assessment)), and biological characteristics (serum albumin level, C-reactive protein (CRP), lymphocyte count and the modified Glasgow Prognostic Score (mGPS)) were retrospectively collected from medical records. The mGPS was calculated from serum CRP and albumin levels and is known to be an independent prognostic factor in oncology [21].

### 2.5. Handgrip Strength Measurement and Dynapenia Definition

HGS was measured with a Jamar hydraulic dynamometer which has already proven its reliability [22]. HGS measurement was performed in all patients at baseline before the administration of the antineoplastic treatment, either during the medical consultation or during the first hospital stay.

The HGS measurement protocol was previously described by Ordan et al. [18]. There were 5 possible handle positions, and position 2 is used in our daily practice. The test was performed with the dominant and non-dominant hand. Patients performed maximal isometric contraction within 3 s in both hands. A verbal motivation was given by the physician to access their best score. After the first measurement, a one-minute break was taken before the second measurement for each hand. The highest value from the four measurements was finally collected.

The HGS test value was defined according to different thresholds. First, initially planned analysis was performed using the newly validated EWGSOP2 criteria for dynapenia (HGS < 27 kg for men and <16 kg for women) [5]. Second, we defined dynapenia using an additional exploratory threshold, as an HGS below the sex-based median of our population. These definitions are designated as original (EWGSOP2) and exploratory, respectively.

### 2.6. Chemotherapy-Induced Dose-Limiting Toxicities (DLT), Dose-Limiting Neurotoxicity (DLN) and All-Grade Chemotherapy-Induced Toxicities

Data on chemotherapy-induced toxicities occurring during the first six months after the initiation of first-line chemotherapy were collected from each patient’s electronic health record. All-grade toxicities during anticancer treatment were also collected (including dose-limiting). Chemotherapy-induced toxicity was graded according to the National Cancer Institute-Common Terminology Criteria for Adverse Events (NCI-CTCAE Version 5.0). Chemotherapy-induced DLT was defined as any toxicity leading to dose reduction (temporary or permanent), treatment delays, or permanent treatment discontinuation. Progressive disease as the cause of treatment discontinuation was not considered DLT. Pre-therapeutic dose adaptation was defined as an initial dose reduction by individual clinical appreciation considering patient profile (age, ECOG PS, organ failure, or malnutrition).

Toxicities were analyzed according to each chemotherapy side-effect profile. Neuropathy was only considered in patients receiving oxaliplatin, cisplatin, and docetaxel; hand-foot syndrome (HFS) and oral mucositis only in patients receiving 5-Fluorouracil (5FU)- or capecitabine-based chemotherapy regimens. Finally, nausea and vomiting were not considered in patients receiving 5FU or gemcitabine alone.

### 2.7. Statistical Analyses

Quantitative variables were expressed as mean and standard deviation or median and interquartile range(s) (IQR) and compared using the non-parametric Kruskal–Wallis test. Qualitative data were described by frequencies and percentages and compared with the Chi-square test or Fisher’s exact test when appropriate. All *p*-values were two-sided, and a *p*-value ≤ 0.05 was considered significant. The tests were performed to compare all-grade and dose-limiting toxicities with the original and the exploratory dynapenia thresholds. An additional multivariate analysis was performed including significant patient characteristics in a stepwise regression multivariate analysis. All data were collected using EpiInfo 7.2.5.0 and analyzed using R Studio (R Core Team, 2022).

## 3. Results

### 3.1. Characteristics of Patients

A total of 322 medical records were screened and 244 patients were included (Figure 1). The characteristics of the included population are described in Table 1.

The median age was 69 (IQR, 59.0–74.0) years and the sex ratio was balanced. Colorectal cancer was the most common primary tumor site (*n* = 105, 43.2%). Eighty-four patients (34.4%) were diagnosed with localized disease, whereas 103 (42.2%) were at a metastatic stage. Forty patients (16.4%) underwent a combination of chemotherapy and biotherapy. Most anticancer drugs were potentially neurotoxic (*n* = 189, 77.4%) and the most frequently received chemotherapy regimen was FOLFOX (infusional and bolus 5FU, leucovorin plus oxaliplatin) (*n* = 96, 39.3%).

### 3.2. Handgrip Strength (HGS)

The mean HGS value was 35.8 ± 8.5 kg for men and 22.8 ± 6.3 kg for women. According to the original EWGSOP2 criteria, 23 patients (9.4%) were defined as dynapenic, including 13 men and 10 women. The median HGS value, defining our exploratory threshold was 34 kg (IQR: 30–41.5) for men and 22 kg (IQR: 19–28) for women. According to our exploratory definition, 107 patients (43.8%) were considered dynapenic, including 57 men and 50 women.

### 3.3. Chemotherapy-Induced DLT

A total of 134 patients (54.9%) experienced chemotherapy-induced DLT. The most frequent DLT was neurotoxicity (*n* = 76, 41.3%). Patients with dynapenia according to the original EWGSOP2 definition were significantly older (*p* < 0.001), with worse ECOG PS (*p* < 0.001) and G8 score (*p* = 0.002), and lower serum albumin levels (*p* = 0.012).

The repartition of DLT according to dynapenia as defined by the original EWGSOP2 criteria is shown in Table 2.

There was no significant association between dynapenia and overall type of DLT. Asthenia (21.7% versus 8.6%, *p* = 0.059) and hand-foot syndrome (HFS) (9.1% versus 1.1%, *p* = 0.075) tended to be a more frequent cause of DLT in patients with dynapenia than in patients without. No association was found between dynapenia and DLN (*p* = 0.786). No additional multivariate analysis was performed for the original HGS threshold due to the limited number of patients diagnosed with dynapenia.

The repartition of DLT according to our exploratory HGS median-based threshold is described in Table 3. Using this definition, patients with exploratory low HGS were significantly older (*p* < 0.001) and had a worse ECOG PS (*p* = 0.006), mGPS score (*p* = 0.020), and G8 score (*p* = 0.050). A significantly higher rate of dose-limiting asthenia was observed in patients with below median-based HGS threshold (15% versus 5.8%, *p* = 0.029). The planned multivariate analysis for the median-based threshold, adjusted on age over 65 years, G8 score, ECOG PS, and mGPS did not show any significant relationship with asthenia (*p* = 0.78) or all DLT combined (*p* = 0.2).

### 3.4. All-Grade Toxicity (Dose-Limiting or Not)

The observed all-grade toxicities according to dynapenia as defined by the original EWGSOP2 criteria are shown in Table 4. Patients with dynapenia experienced more HFS (18.2% versus 3.2%, *p* = 0.007) and tended to experience more grade 3–4 diarrhea (25% versus 10%, *p* = 0.071) (Appendix A).

The observed all-grade toxicities according to our exploratory median-based HGS threshold are shown in Table 5. Patients with exploratory low HGS experienced more anemia (77.6% versus 59.9%, *p* = 0.006), more asthenia (97.2% versus 87.6% *p* = 0.014), and less vomiting (18.1% versus 30.3%, *p* = 0.047).

## 4. Discussion

In the present study, pre-therapeutic dynapenia was not associated with chemotherapy-induced DLT in digestive cancer patients receiving first-line chemotherapy. Patients with dynapenia, as defined by EWGSOP2, seemed to experience more HFS and serious diarrhea. However, the current threshold used to define dynapenia is not consensual, especially in cancer patients [11]. Consequently, we performed an exploratory analysis based on the median HGS of our population. Using this new threshold to define dynapenia, we observed more dose-limiting asthenia and anemia.

The prevalence of DLT was 54.9% in this study. Previous studies observed similar DLT rates [20,23]. Botsen et al. reported 49% of chemotherapy-induced DLT in digestive cancer patients [20]. Celik et al. observed 52% of DLT in digestive cancer patients receiving platinum-based chemotherapy [23]. He also described higher rates (78.9%) in patients with sarcopenia (defined as a low muscle mass measured on a computed tomography), suggesting its potential role in predicting chemotherapy-induced DLT. Nonetheless, HGS alone seemed to be insufficient to predict the occurrence of DLT [23]. In another study, Lakenman et al. showed an association between low HGS and DLT during neoadjuvant chemoradiation in patients with esophageal cancer [24]. In this study, dynapenia was defined below the tenth percentile (HGS < 37.6 kg for men and <23.6 kg for women) [24].

Conversely to the FIGHTDIGO study, and despite a larger sample without a selection bias of non-neurotoxic treatment, we did not find any association between pre-therapeutic dynapenia and DLN [20]. Because dynapenic patients were less exposed to major neuropathic-providing chemotherapies such as docetaxel and cisplatin, they could experience less DLN.

Indeed, patients with exploratory low HGS were significantly older and had a worse ECOG PS, mGPS score, and geriatric G8 score. Our results are in line with the known association of low HGS with markers of functional and nutritional status [18,25], age [26], and geriatric G8 score [27]. These findings support the usefulness of HGS measurement as an interesting additional tool to identify frailty in cancer patients.

In the exploratory analysis, in which low HGS was defined by a HGS value below the sex-based median, patients experienced more dose-limiting asthenia in univariate analysis. Kilgour et al. have already observed more fatigue in patients with weaker muscle strength [14]. Additionally, asthenia could be a part of the cachexia syndrome [28], which is defined as a metabolic syndrome associated with an underlying chronic disease and characterized by a loss of skeletal muscle mass [6]. Cachexia is generally associated with chronic inflammation [29,30,31]. Sarcopenia, cachexia, and asthenia share common and overlapping characteristics. Our findings might be an additional item for the interaction between these different clinical entities. HGS could be used for the assessment of sarcopenia in daily practice as a part of the spectrum of cachexia [5]. However, due to low statistical power, the higher risk of a false positive association should be taken into account. Despite the absence of a strong statistical association, dynapenia seems to be a part of a larger frailty syndrome.

In the same analysis, patients with exploratory low HGS tend to experience more frequent all-grade anemia but not DLT. Previous studies have shown higher hematological toxicity in patients with lower HGS and sarcopenia [32,33]. However, this toxicity is commonly managed by blood transfusion support and/or erythropoiesis-stimulating agents before treatment’s dose adaptation or delay [34,35]. Cancer-induced inflammation inhibits hematopoiesis by the interaction of interleukin 6 and tumor necrosis factor-alpha (TNF-α) [36]. However, the role of cachexia remains unclear. In a prospective study, Rocha et al. described an increased risk of grade 3–4 toxicities in the first three cycles of chemotherapy in patients with cachexia treated for gastrointestinal cancer [37]. However, their follow-up was shorter in comparison with our study, and potential consequences of cachexia on erythropoiesis have not been detected due to the lifespan of blood cells. In non-cancer populations, the association between anemia and low HGS has already been described [38]. Our results suggest that HGS could be interesting to improve the management of cancer-related anemia and be a part of the adaptation of supportive care.

Patients with dynapenia also experienced more HFS (*p* = 0.007). This result relies on a very low number of patients and should be interpreted with caution. Risk factors of HFS have been previously described, including age, sex, and genetic susceptibilities [39,40]. To our knowledge, no association between muscle strength and HFS has been previously observed. Gökyer et al. described higher DLT and HFS rates in the sarcopenic population with colorectal cancer receiving regorafenib [41]. Although, HGS was not measured [41].

Patients with dynapenia as defined by our exploratory thresholds experienced less vomiting without DLT. This group had fewer chemotherapy combinations (such as FOLFIRINOX or TFOX) which are associated with more vomiting [42]. However, the population with exploratory low HGS was older, whereas younger patients are at greater risk of chemotherapy-induced nausea and vomiting [43].

Sarcopenia had already shown promising results in predicting chemotherapy-related toxicity but adapted criteria for assessment are needed [11]. Low lean body mass was associated with the increased occurrence of chemotherapy-related toxicities [44]. The relationship between body composition and pharmacokinetics of chemotherapy is also well established [45]. Recently, Cereda et al. demonstrated that muscle weakness was a better predictor of survival than skeletal muscle mass estimated by bioelectrical impedance analysis [46].

Heterogeneous cut-off points have been previously described to define dynapenia [9,10,17,24,47]. Nowadays, the gold standard is the EWGSOP2 definition, based on Dodds et al. study [9]. However, most of these thresholds (including the current gold standard) have been established in non-oncologic geriatric populations. Thus, this lack of consensus is an issue that still needs to be addressed as it hampers comparisons between studies and the emergence of new guidelines for daily oncology practice. Indeed, the consequences of cancer on muscle strength are not included in the used definitions. Therefore, an exploratory cut-off point was assessed in our study, but further prospective studies are needed to validate its relevance. A recent study based on 6182 patients found that HGS cut-offs of <36 kg for men and <23 kg for women were the best ones to predict mortality in the elderly [48]. Their new thresholds are similar to those defined by Lakenman et al. in oesophageal cancer patients and in our exploratory analysis [24,48].

The present study had several limitations. First, the study was based on a retrospective examination of medical records which limits the exhaustive collection of every toxicity. However, our oncology unit has a strong culture of grading and tracing every toxicity. Moreover, chemotherapy was prescribed on a unique software limiting selection bias. Second, some comorbidities influencing DLT were not recorded, such as heart failure or renal insufficiency. The use of the Charlson Comorbidity Index could have been useful to prevent a possible confusion bias. Third, the included population was heterogeneous with various types of digestive cancers and chemotherapy regimens. Fourth, we observed a very limited number of DLT, hindering the pre-planned multivariate analysis that could have helped us better understand the interaction between potential confusion factors. This study also presented several strengths including the analysis of a large cohort of outpatients with digestive cancer and providing real-life daily-practice data.

HGS measurement with the JAMAR dynamometer is an easy-to-use, portable and economical way to screen for dynapenia in daily clinical practice [19]. Further studies could focus on the prevention of potential toxicities, the evolution of HGS throughout the anticancer treatment program, or the usefulness of Adapted Physical Activity (APA) programs in sarcopenic patients. Muscle strength follow-up could be used in daily practice during APA programs. Recently, APA has been reported as feasible in cancer outpatients beginning medical anticancer treatment [49]. Indeed, recent studies have shown improvement in quality of life and a reduction in fatigue through physical activity [50].

## 5. Conclusions

In conclusion, digestive cancer outpatients with pre-therapeutic dynapenia, according to EWGSOP2 criteria, do not seem to have more chemotherapy-induced DLT. Based on an exploratory higher cut-off point, low HGS could be a predictive factor of chemotherapy-induced anemia and asthenia. There is a growing need to better define the HGS cut-off points of dynapenia in cancer patients. The HGS measurement is easily use in daily practice, non-invasive and inexpensive. The diagnosis of dynapenia could help the care provider to better assess patients’ frailty, and to adjust nutritional care and APA before the appearance of chemotherapy-induced toxicities.

## Figures and Tables

**Figure 1 nutrients-14-04448-f001:**
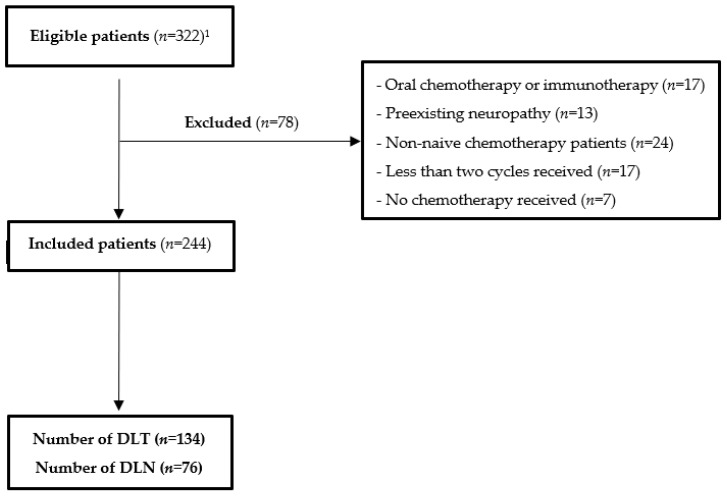
Flow chart of the FIGHTDIGOTOX study. Abbreviations: DLN: dose-limiting neurotoxicity; DLT: dose-limiting toxicity; UMA-CH, ambulatory cancer unit. ^1^ 69 patients were assessed in AgElOn.

**Table 1 nutrients-14-04448-t001:** Overall population characteristics and according to dynapenia (EWGSOP2 criteria).

Characteristics of Patients	Level	Overall	Dynapenia ^1^	Normal HGS ^1^	*p*-Value
Total, *n* (%)		244	23 (9.4)	221 (90.6)	
Sex, *n* (%)	Female Male	109 (44.7) 135 (55.3)	10 (43.5) 13 (56.5)	99 (44.8) 122 (55.2)	1.000
Age, median (IQR)		69.0 (59.0–74.0)	73.0 (69.0–81.5)	68.0 (58.0–73.0)	<0.001
BMI, median (IQR)		24.6 (21.5–28.6)	24.6 (21.1–29.4)	24.6 (21.6–28.6)	0.862
ECOG PS, *n* (%)	0 1 2 3	66 (27.0) 150 (61.5) 26 (10.7) 2 (0.8)	0 (0.0) 14 (60.9) 8 (34.8) 1 (4.3)	66 (29.9) 136 (61.5) 18 (8.1) 1 (0.5)	<0.001
Serum albumin level, median (IQR)		39.0 (36.0–42.0)	36.0 (33.0–40.0)	39.0 (37.0–42.0)	0.012
CRP, median (IQR)		9.0 (4.0–33.2)	19.0 (4.5–40.5)	8.0 (4.0–33.0)	0.114
mGPS, *n* (%)	0 1 2	118 (48.4) 87 (35.7) 39 (16.0)	5 (21.7) 10 (43.5) 8 (34.8)	113 (51.1) 77 (34.8) 31 (14.0)	0.006
Lymphopenia, *n* (%)	No Yes	226 (92.6) 18 (7.4)	21 (91.3) 2 (8.7)	205 (92.8) 16 (7.2)	0.681
G8 score, median ^2^ (IQR)		12.0 (11.0–15.0)	10.0 (8.8–12.0)	13.0 (11.0–15.0)	0.002
Primary tumor location, *n* (%)	Colon and rectum Stomach Esophagus Pancreas Others ^3^	105 (43.2) 26 (10.7) 18 (7.4) 69 (28.4) 25 (10.2)	16 (69.6) 2 (8.7) 2 (8.7) 2 (8.7)	89 (40.5) 24 (10.9) 16 (7.3) 67 (30.5) 25 (11.1)	0.339
Stage, *n* (%)	Localized Locally advanced Metastatic	84 (34.4) 57 (23.4) 103 (42.2)	9 (39.1) 4 (17.4) 10 (43.5)	75 (33.9) 53 (24.0) 93 (42.1)	0.784
Number of metastatic sites, *n* (%)	1 ≥2	68 (65.4) 36 (34.6)	7 (70.0) 3 (30.0)	61 (64.9) 33 (35.1)	1.000
Chemotherapy regimen, *n* (%)	5FU + Oxaliplatin 5FU + Irinotecan + Oxaliplatin 5FU alone Gemcitabine Others ^4^	96 (39.3) 69 (28.3) 24 (9.8) 18 (7.4) 37 (15.1)	12 (52.2) 3 (13.0) 6 (26.1) 2 (8.6)	84 (38.0) 66 (29.9) 18 (8.1) 18 (8.1) 35 (16.2)	0.170
Biotherapy, *n* (%)	None Bevacizumab Others ^5^	204 (83.6) 26 (10.7) 14 (5.7)	18 (78.3) 5 (21.7)	186 (84.2) 21 (9.5) 14 (6.3)	0.305
Concomitant radiotherapy, *n* (%)	No Yes	222 (91.0) 22 (9.0)	22 (95.7) 1 (4.3)	200 (90.5) 21 (9.5)	0.704

Abbreviations: 5FU: 5 Fluorouracil; BMI: Body Mass Index; CRP: C-reactive protein; ECOG PS: Eastern Cooperative Oncology Group Criteria Performance Status; HGS: handgrip strength; IQR: interquartile range; mGPS: modified Glasgow prognosis score. ^1^ According to the EWGSOP2 definition; ^2^ Data available for 82 patients; ^3^ Other localizations: biliary tract (*n* = 8), small intestine (*n* = 7), ampulla of Vater (*n* = 3), neuroendocrine tumor (*n* = 4), appendix (*n* = 1), anal (*n* = 1), unknown primary (*n* = 1); ^4^ Other chemotherapy: 5FU + Irinotecan (*n* = 8), 5FU + Oxaliplatin + Docetaxel (*n* = 9), 5FU + Cisplatin (*n* = 1), 5FU + Dacarbazine (*n* = 3), Carboplatin-Etoposide (*n* = 1), Gemcitabine + Cisplatin (*n* = 4), Gemcitabine + Oxaliplatin (*n* = 3), Capecitabine + Oxaliplatin (*n* = 7), Capecitabine + Mitomycin (*n* = 1); ^5^ Other biotherapy: Panitumumab (*n* = 10), Trastuzumab (*n* = 4).

**Table 2 nutrients-14-04448-t002:** Association between dynapenia (original EWGSOP2 criteria) and chemotherapy-induced dose-limiting toxicity (DLT).

Dose Limiting Toxicity	Overall (*n* = 244)	Dynapenia ^1^ (*n* = 23)	Normal HGS ^1^ (*n* = 221)	*p* Value
All Type (%)	134 (54.9)	13 (56.5)	121 (54.8)	1.000
Neuropathy ^2^	76 (41.3)	7 (46.7)	69 (40.8)	0.786
Asthenia (%)	24 (9.8)	5 (21.7)	19 (8.6)	0.059
Diarrhea (%)	20 (8.2)	2 (8.7)	18 (8.1)	1.000
Nausea ^3^ (%)	4 (2.0)	1 (5.9)	3 (1.6)	0.298
Vomiting ^3^ (%)	4 (2.0)	0 (0)	4 (2.2)	1.000
Neutropenia (%)	28 (11.5)	0 (0)	28 (12.7)	0.086
Anemia (%)	6 (2.5)	1 (4.3)	5 (2.3)	0.451
Thrombopenia (%)	13 (5.3)	2 (8.7)	11 (5.0)	0.352
Hand foot syndrome ^4^ (%)	4 (1.9)	2 (9.1)	2 (1.1)	0.075
Oral mucositis ^4^ (%)	3 (1.2)	0 (0)	3 (1.4)	1.000

Abbreviations: DLT; dose-limiting toxicity; HGS: handgrip strength. ^1^ According to the EWGSOP2 definition; ^2^ Only patients receiving neurotoxic chemotherapy (*n* = 184); ^3^ Patients receiving 5FU and gemcitabine alone were not analyzed for this adverse effect (*n* = 202); ^4^ Only patients receiving 5FU- or capecitabine-based chemotherapy regimen (*n* = 210).

**Table 3 nutrients-14-04448-t003:** Association between low handgrip strength on median-based analysis (exploratory low HGS) and chemotherapy-induced dose-limiting toxicity (DLT).

Dose Limiting Toxicity	Overall (*n* = 244)	Exploratory Low HGS ^1^ (*n* = 107)	Normal HGS ^1^ (*n* = 137)	*p*-Value for Univariate Analysis	*p*-Value for Multivariate Analysis *
All Type (%)	134 (54.9)	32 (29.9)	40 (29.2)	1.000	0.2
Neuropathy ^2^	76 (41.3)	26 (36.1)	50 (44.6)	0.285	-
Asthenia (%)	24 (9.8)	16 (15.0)	8 (5.8)	0.029	0.78
Diarrhea (%)	20 (8.2)	9 (8.4)	11 (8.0)	1.000	-
Nausea ^3^ (%)	4 (2.0)	2 (2.4)	2 (1.7)	1.000	-
Vomiting ^3^ (%)	4 (2.0)	1 (1.2)	3 (2.5)	0.645	-
Neutropenia (%)	28 (11.5)	10 (9.3)	18 (13.1)	0.421	-
Anemia (%)	6 (2.5)	4 (3.7)	2 (1.5)	0.409	-
Thrombopenia (%)	13 (5.3)	5 (4.7)	8 (5.8)	0.779	-
Hand foot syndrome ^4^ (%)	4 (1.9)	2 (2.2)	2 (1.7)	1.000	-

Abbreviations: DLT: dose-limiting toxicity; HGS: handgrip strength; mGPS: modified Glasgow prognosis score. ^1^ HGS cut-off based on the median in the population as HGS < 34 kg for men and <22 kg for women; ^2^ Only patients receiving neurotoxic chemotherapy (*n* = 184); ^3^ Patients receiving 5FU and gemcitabine alone were not analyzed for this adverse effect (*n* = 202); ^4^ Only patients receiving 5FU- or capecitabine-based chemotherapy regimen (*n* = 210); * Multivariate analysis was adjusted on age over 65 years, performance status, G8 score, and mGPS; -: statistical analysis not performed due to futility.

**Table 4 nutrients-14-04448-t004:** Association between dynapenia (original EWGSOP2 criteria) and all-grade chemotherapy-induced toxicities (dose-limiting and not).

Toxicity (All Grade)	Overall (*n* = 244)	Dynapenia ^1^ (*n* = 23)	Normal HGS ^1^ (*n* = 221)	*p* Value
Neuropathy ^2^ (%)	174 (94.6)	14 (93.3)	160 (94.7)	0. 582
Asthenia (%)	224 (91.8)	23 (100.0)	201 (91.0)	0.303
Diarrhea (%)	139 (57.0)	12 (52.2)	127 (57.5)	0.693
Nausea ^3^ (%)	115 (56.9)	9 (52.9)	106 (57.3)	0.801
Vomiting ^3^ (%)	51 (25.2)	3 (17.6)	48 (25.9)	0.605
Neutropenia (%)	60 (24.6)	3 (13.0)	57 (25.8)	0.286
Anemia (%)	165 (67.6)	19 (82.6)	146 (66.1)	0.238
Thrombopenia (%)	73 (29.9)	10 (43.5)	63 (28.5)	0.235
Hand foot syndrome ^4^ (%)	10 (4.8)	4 (18.2)	6 (3.2)	0.007
Oral mucositis ^4^ (%)	29 (11.9)	2 (8.7)	27 (12.2)	0.836

Abbreviations: DLT: dose-limiting toxicity; HGS: handgrip strength. ^1^ According to the EWGSOP2 definition; ^2^ Only patients receiving neurotoxic chemotherapy (*n* = 184); ^3^ Patients receiving 5FU and gemcitabine alone were not analyzed for this adverse effect (*n* = 202); ^4^ Only patients receiving 5FU- or capecitabine-based chemotherapy regimen (*n* = 210).

**Table 5 nutrients-14-04448-t005:** Association between low handgrip strength on median-based analysis (exploratory low HGS) and all-grade chemotherapy-induced toxicities (dose-limiting and not).

Toxicity (All Grade)	Overall (*n* = 244)	Exploratory Low HGS ^1^ (*n* = 107)	Normal HGS ^1^ (*n* = 137)	*p* Value
Neuropathy ^2^ (%)	174 (94.6)	66 (91.7)	108 (96.4)	0.193
Asthenia (%)	224 (91.8)	104 (97.2)	120 (87.6)	0.014
Diarrhea (%)	139 (57.0)	56 (52.3)	83 (60.6)	0.214
Nausea ^3^ (%)	115 (56.9)	41 (49.4)	74 (62.2)	0.084
Vomiting ^3^ (%)	51 (25.2)	15 (18.1)	36 (30.3)	0.047
Neutropenia (%)	60 (24.6)	26 (24.3)	34 (24.8)	1.000
Anemia (%)	165 (67.6)	83 (77.6)	82 (59.9)	0.006
Thrombopenia (%)	73 (29.9)	32 (29.9)	41 (29.9)	1.000
Hand foot syndrome ^4^ (%)	10 (4.8)	5 (5.4)	5 (4.3)	0.629
Oral mucositis ^4^ (%)	29 (11.9)	15 (14.0)	14 (10.2)	0.455

Abbreviations: DLT: dose-limiting toxicity; HGS: handgrip strength. ^1^ HGS cut-off based on the median in the population as HGS <34 kg for men and <22 kg for women; ^2^ Only patients receiving neurotoxic chemotherapy (*n* = 184); ^3^ Patients receiving 5FU and gemcitabine alone were not analyzed for this adverse effect (*n* = 202); ^4^ Only patients receiving 5FU- or capecitabine-based chemotherapy regimen (*n* = 210).

## Data Availability

The data presented in this study are available on request from the corresponding author.

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
