# Peer review of "Association of Low Handgrip Strength with Chemotherapy Toxicity in Digestive Cancer Patients: A Comprehensive Observational Cohort Study (FIGHTDIGOTOX)"

_nutrients, 2022, doi:10.3390/nu14214448_

Round 1
Reviewer 1 Report
This study aimed to assess the relationship between pre-therapeutic hand-grip strength (HGS) and chemotherapy-induced dose-limiting toxicity (DLT) and/or all-grade toxicity in digestive 76 cancer patients treated in an outpatient cancer unit.
The authors conclude that pre-therapeutic dynapenia wasn’t a predictive factor for overall DLT and neurotoxicity in digestive cancer patients but could be a predictive factor of chemotherapy-induced anemia and asthenia.
COMMENTS
The message of this investigation is modest but is coherent with the findings of the research.
My comments regard some statistical issues.
1. Since patients who were classified “with dynamopenia” were also significantly older and with low performance status than control group, two conditions which are known to decrease the tolerance to chemotherapy, it is difficult to attribute the poorer compliance to dynaopenia without a multivariable analysis which includes most of variables which result significantly different in the two groups at the baseline.
2. Statistical significance in tables 2 to 5 should not be corrected by the Bonferroni factor?
Author Response
The message of this investigation is modest but is coherent with the findings of the research. My comments regard some statistical issues.
Point 1. Since patients who were classified “with dynamopenia” were also significantly older and with low performance status than control group, two conditions which are known to decrease the tolerance to chemotherapy, it is difficult to attribute the poorer compliance to dynaopenia without a multivariable analysis which includes most of variables which result significantly different in the two groups at the baseline.
Authors: We would like to thank the reviewer for pointing out this potential confusion bias.
Indeed, our univariate analyses showed an association between dynapenia using the EWGSOP2 definition and age, albumin, mGPS, ECOG PS individually.
However, due to the low number of patients with dynapenia (n = 15) and DLT with the EWGSOP2 definition (n = 23), a multivariate analysis adding 5 covariates would violate the "one-in-ten" rule. This rule specifies that one additional covariate can be added to the logistic regression model every ten events. The number of event (DLT) is therefore insufficient to safely run a model as ambitious as proposed, without taking a risk of withdrawing false conclusions.
We encountered the same problem for the exploratory threshold (dynapenia : n = 107 patients, DLT : n = 32 events).
In line with this comment, you can already read line 225-227: "No additional multivariate analysis was performed for the original HGS threshold due to the limited number of patients diagnosed with dynapenia."
To emphasize this argument, we also added to the discussion section, line 360-362: "Fourth, we observed a very limited number of DLT, hindering the pre-planned multivariate analysis that could have helped us better understand the interaction between potential confusion factors."
Point 2. Statistical significance in tables 2 to 5 should not be corrected by the Bonferroni factor?
Authors: We thank you for underlining this potential issue.
The Bonferroni technique is an adjustment technique which can be used to minimize the risk of type 1 error (false positive) when multiple independent comparisons are performed.
For table 2 and 5, ten independent comparisons have been made, in each table.
For ? = 0.05 as defined in our methods and n = 10 as the number of comparisons, the new alpha threshold ?′ = ?/n = 0.005. By using ?′ as the new threshold for significance, hand-foot syndrome (HFS) (p = 0.007, Table 2) and asthenia (p = 0.014, Table 5) would not be considered as statistically significant anymore.
However, we would like to underline the fact that the higher relative frequency of HFS in the low HGS group relies on a low number of events. Moreover, the absolute difference of patients for asthenia in Table 5 is also low (16 patients, 7.1% of the total of patients with asthenia). This lack of statistical power will not be corrected by the Bonferroni technique. Alternatively, we propose to underline this fact in our manuscript to prevent the readership of the journal from misinterpreting the results of the study.
To underline this lack of statistical power, we added to the discussion section about asthenia line 305-306: "However, due to low statistical power, the higher risk of a false positive association should be taken into account" and HFS line 323-324: "This result rely on a very low number of patients and should be interpreted with caution",
and see our answer to first comment, we also added to the discussion section, line 360-362: "Fourth, we observed a very limited number of DLT, hindering the pre-planned multivariate analysis that could have helped us better understand the interaction between potential confusion factors."

Reviewer 2 Report
The current manuscript intends to explore the utility of substitutionary measures such as handgrip strength to measure levels of sarcopenia and dose-limiting neurotoxicities instead of the traditionally accepted costlier, invasive, and irradiational measures. This study shows adequate scientific interest and merit because it explores the efficacy of a simplistic approach to a rather challenging problem. The sample size appears adequate. The paper appears to be neatly organized and well written. There is only one suggestion:
1) The conclusion needs to expand on the implications of the study findings in terms of benefit to the patients, care providers, and the system at large.
Author Response
The current manuscript intends to explore the utility of substitutionary measures such as handgrip strength to measure levels of sarcopenia and dose-limiting neurotoxicities instead of the traditionally accepted costlier, invasive, and irradiational measures. This study shows adequate scientific interest and merit because it explores the efficacy of a simplistic approach to a rather challenging problem. The sample size appears adequate. The paper appears to be neatly organized and well written. There is only one suggestion:
1) The conclusion needs to expand on the implications of the study findings in terms of benefit to the patients, care providers, and the system at large.
Authors: We would like to thank you very much for your review. To follow your recommendation, we propose to add the following sentence to the conclusion section line 378-381:
“The HGS measurement is easily use in daily practice, non-invasive and inexpensive. Diagnosis of dynapenia could help the care provider to better assess patients' frailty and to adjust nutritional care and APA before the appearance of chemotherapy-induced toxicities.”
